# Genome-Wide Analysis of the bHLH Gene Family in *Loropetalum chinense* var. *rubrum*: Identification, Classification, Evolution, and Diversity of Expression Patterns under Cultivation

**DOI:** 10.3390/plants12193392

**Published:** 2023-09-26

**Authors:** Yang Liu, Ling Lin, Yang Liu, Qiong Mo, Damao Zhang, Weidong Li, Xingyao Xiong, Xiaoying Yu, Yanlin Li

**Affiliations:** 1College of Horticulture, Engineering Research Center for Horticultural Crop Germplasm Creation and New Variety Breeding (Ministry of Education), Hunan Mid-Subtropical Quality Plant Breeding and Utilization Engineering Technology Research Center, Hunan Agricultural University, Changsha 410128, China; liuyang1203@stu.hunau.edu.cn (Y.L.); 418272846@stu.hunau.edu.cn (Y.L.); momo25@stu.hunau.edu.cn (Q.M.); zdm1558@stu.hunau.edu.cn (D.Z.); 2School of Economics, Hunan Agricultural University, Changsha 410128, China; lljjxy@hunau.edu.cn; 3Hunan Key Laboratory of Germplasm Innovation and Comprehensive Utilization of Garden Flowers, Hunan Horticulture Research Institute, Hunan Academy of Agricultural Sciences, Changsha 410125, China; liweidong@hunaas.cn; 4Agricultural Genomics Institute at Shenzhen, Chinese Academy of Agricultural Sciences, Shenzhen 518120, China; 5Kunpeng Institute of Modern Agriculture, Foshan 528225, China; 6School of Biological Sciences, Nanyang Technological University, 60 Nanyang Drive, Singapore 637551, Singapore

**Keywords:** *L. chinense* var. *rubrum*, bHLH transcription factor, expression analyses, anthocyanin

## Abstract

The basic helix–loop–helix (bHLH) transcription factor family is the second-largest transcription factor family in plants. Members of this family are involved in the processes of growth and development, secondary metabolic biosynthesis, signal transduction, and plant resistance. *Loropetalum chinense* var. *rubrum* is a critical woody plant with higher ornamental and economic values, which has been used as ornamental architecture and traditional Chinese herbal medicine plants. However, the bHLH transcription factors in *Loropetalum chinense* var. *rubrum* (*L. chinense* var. *rubrum*) have not yet been systematically demonstrated, and their role in the biosynthesis of anthocyanin is still unclear. Here, we identified 165 potential *LcbHLHs* genes by using two methods, and they were unequally distributed on chromosomes 1 to 12 of the genome of *L. chinense* var. *rubrum*. Based on an evolutionary comparison with proteins from *Arabidopsis* and *Oryza sativa*, these bHLH proteins were categorized into 21 subfamilies. Most LcbHLHs in a particular subfamily had similar gene structures and conserved motifs. The Gene Ontology annotation and Cis-elements predicted that *LcbHLHs* had many molecular functions and were involved in processes of plant growth, including the biosynthesis of flavonoids and anthocyanins. Transcriptomic analysis revealed different expression patterns among different tissues and cultivars of *L. chinense* var. *rubrum*. Many *LcbHLHs* were expressed in the leaves, and only a few genes were highly expressed in the flowers. Six *LcbHLHs* candidate genes were identified by bioinformatics analysis and expression analysis. Further Real-time quantitative PCR analysis and protein interaction network analysis showed that LcbHLH156, which is one of the candidate proteins belonging to the IIIf subfamily, could interact with proteins related to anthocyanin synthesis. Therefore, *LcbHLH156* was transiently expressed in *L. chinense* var. *rubrum* to verify its function in regulating anthocyanin synthesis. Compared with the control group, red pigment accumulation appeared at the wound after injection, and the total anthocyanin content increased at the wound of leaves. These results lay a foundation for the research of the regulation mechanism of leaf colors in *L. chinense* var. *rubrum* and also provide a basis for the function of the LcbHLH family.

## 1. Introduction

Transcription factors (TFs) often serve as a central regulator to regulate the expression of target genes. They can form intricate networks through protein–protein interactions to control or affect many biological processes at the transcription level [1]. Among the various TFs, the bHLH family is a relatively large family of transcription factors widely found in diverse eukaryotes. The bHLH domain is often made up of 50–60 amino acids and has two functionally distinct regions: a length of 10–15 basic amino acids (the basic region) and a stretch of roughly 40 amino acids (the helix–loop–helix region). The helix–loop–helix region contains two amphipathic α-helices separated by an intervening loop [2,3]. The basic region is located at the N-terminus of the bHLH domain and is the DNA-binding region that allows the bHLH TFs to bind to the E-box (CANNTC). In contrast, the HLH region is located at the C-terminus of the bHLH domain. It acts as a dimerization domain, which can promote the formation of homodimers or heterodimers between proteins to alter the expression of target genes involved in various signaling pathways [4,5]. In the bHLH protein domain, only 19 amino acids are conserved, and it possesses a highly conservative H–E–R motif. Previous studies have typically classified members of the bHLH superfamily into subfamilies or subgroups based on the conserved motifs, evolutionary relationships, and structural domains. Ledent and Atchley classified the bHLH transcription factors of animals into six subgroups (A–F) based on their sequence homology and evolutionary relationships [3,6]. However, there are differences in classification between plants and animals because of the relative independence of the bHLH transcription factors’ genealogy. So far, no definite categories of the families of plant bHLH transcription factors have been proposed [7]. Generally, the bHLH transcription factor families in plants have been divided into 15–26 groups [3,8,9]. For example, 147 and 167 bHLH family members were identified from *Arabidopsis* and *Oryza sativa*, respectively, and then classified into 24 and 23 subfamilies [9,10]. Subsequently, the various functions of the *bHLH* gene have also been validated.

Through previous studies, scientists have classified the function of *bHLH* genes in *Arabidopsis* and found that *bHLH* genes play different roles in different aspects [11]. In terms of the mineral nutrition of plants, *AtbHLH121*, *AtbHLH18*, *AtbHLH19*, *AtbHLH20*, and *AtbHLH25* can regulate iron homeostasis by indirectly activating the FER–LIKE IRON DEFICIENCY–INDUCED TRANSCRIPTION FACTOR (FIT) [12]. In growth and development, *AtLP1* and *AtLP2* can regulate longitudinal cell elongation [13]. Moreover, *AtNFL* promotes flowering under short-day conditions in *Arabidopsis* [14]. The *PobHLH5* and *PobHLH8* of *P. ostreatus* infestation in *Arabidopsis* verified that *bHLH* genes play an important role in growth and development [15]. In addition, *bHLHs* are also involved in plants’ abiotic stress responses, signal transmission, secondary metabolism, etc. [16]. For example, the overexpression of *TabHLH39* improved drought tolerance, salt tolerance, and frost resistance in transgenic *Arabidopsis* [17], and *MdCIB1* in apples plays an active role in drought resistance [18]. Researchers have also found that the expression of *CpbHLH1* in transgenic model plants suppressed the accumulation of anthocyanin [19], and *NnTT8* in the lotus has been indicated to be involved in the positive regulation of the biosynthesis of anthocyanin [20]. It has been further shown that pomegranate fruits’ anthocyanins can be regulated by a combination of bHLH and MYB [21], particularly the formation of the MYB–bHLH–WD40 ternary complex [22,23,24].

*Loropetalum chinense* var. *rubrum* (*L. chinense* var. *rubrum*) is a plant with characteristic vivid flowers and foliage found in Asia, Europe, and America. Ornamental plant lovers deeply enjoy its beautiful foliage and pretty shape [25]. The ornamental value of *L. chinense* var. *rubrum* mainly includes its leaf color, flower color, and tree shape. Among which, leaf color is one of the essential ornamental values. Previous studies have shown that anthocyanin is a crucial compound for the leaf color of plants [26,27]. Its synthesis is usually affected by the external environment, synthetic structural genes, and regulatory genes [28]. High temperature and drought usually cause the regreening of *L. chinense* var. *rubrum* leaves mainly due to anthocyanin degradation [29,30]. In addition, researchers have also further explored the coloration mechanism of *L. chinense* var. *rubrum* leaves at the molecular level and have identified and cloned structural genes related to the synthesis of anthocyanin, including *LcCHI* and *LcANS* [31]. The *bHLH* gene is one of the essential regulatory genes in the anthocyanin synthesis pathway, which can encode the corresponding transcription factors, and then activate or inhibit the spatial and temporal expression of structural genes by forming the MYB–bHLH–WD40 complex, thereby regulating the synthesis of plant anthocyanins [24,32]. It has been reported that the genome sequencing and identification of the bHLH family have been completed in various plants, such as Arabidopsis, potato, bamboo, grape, etc. [11,33,34,35]. However, the characteristics of the bHLH transcription factors and their roles in *L. chinense* var. *rubrum* are still unclear.

In this study, the bHLH gene family of *L. chinense* var. *rubrum* was systematically identified using bioinformatic methods. The potential *bHLH* genes regulating leaf color were filtered by a series of biological analyses and expression patterns. The function of *LcbHLH156* in anthocyanin synthesis was preliminarily verified by transient expression. These results provided us a further understanding of the structure, function, and evolution of the bHLH family in *L. chinense* var. *rubrum*, and they may provide a potential basis for exploring the regulatory network of leaf coloration in *L. chinense* var. *rubrum*.

## 2. Results

### 2.1. Identification of the LcbHLHs and Evolutionary Analysis

In total, 165 bHLH members were acquired via the method of homologous BLAST and HMMER searches of the genomic data of *L. chinense* var. *rubrum*, which were named LcbHLH1 to LcbHLH165 according to their chromosomal localization (Figure 1, Appendix A). Predictions of the physicochemical properties by using ExPasy revealed that the number of amino acids contained in the bHLH protein sequence of *L. chinense* var. *rubrum* ranged from 120 (LcbHLH96) to 1186 aa (LcbHLH164), with an average of 375 aa. The molecular weight of these proteins ranged from 13,460.46 (LcbHLH96) to 130,429.73 Da (LcbHLH84), with an average of 41,508.43 Da, and the theoretical isoelectric points ranged from 4.70 (LcbHLH62, LcbHLH102, and LcbHLH103) to 10.12 (LcbHLH96), with 62.71% of them with isoelectric points lower than 7, which were predicted to be acidic. These results were consistent with previous studies of the patterns of the isoelectric points in *Arabidopsis* and *Oryza sativa*. The grand average of hydropathicity (GRAVY) of the proteins was from −1.037 to −0.102, showing that all LcbHLHs are hydrophilic. The instability index (II) ranged from 36.68 to 71.31, with only two proteins shown to be stable (II < 40), and the aliphatic index was between 51.42 and 105.58 (Appendix A). Most of the LcbHLHs were localized in the nucleus, and only a few were distributed in the cytoplasm, chloroplasts, plasmodesmata, and Golgi apparatus. No signal peptide was found for any of the LcbHLHs using SignalP, indicating that they are non-secretory proteins.

To clarify the evolutionary relationships of *L. chinense* var. *rubrum* with Arabidopsis (*Arabidopsis thaliana*) and rice (*Oryza sativa*), an evolutionary bHLH proteins tree was constructed (Appendix A). Through the clustering of the evolutionary tree, we found that the bootstrap value of the root node is small. As the evolution progresses, the bootstrap value between the same subfamilies gradually increases. According to the previous classification criteria and a clustering analysis of the evolutionary tree in Arabidopsis and rice, 24 subfamilies of bHLH TFs were identified. *L. chinense* var. *rubrum* has 21 subfamilies, and only three subfamilies (XIII, XIV, and XV) contained no members (Figure 2). With 20 members, Subfamily XII was the largest subfamily of LcbHLHs, while Subfamilies IVd, VI, and X were the smallest, with only one member. In addition, compared with *Arabidopsis* [5], *Ficus carica* L. [36], and *Oryza sativa* [7,10,37], Subfamilies III(a + b + c), III(d + e), the IVa, VII(a + b), and XII were considerably expanded (Table 1).

### 2.2. Multiple Sequence Alignment and Analyses of the Motifs, Domains, and Structure

To further elucidate the structural features of LcbHLH proteins, a multiple sequence alignment was performed on the structural domains of bHLH. As shown in Figure 3, each LcbHLH protein displayed four conserved regions, including one basic region, one loop region, and two helix regions. Most LcbHLH proteins possessed highly conserved basic regions and two helixes, except for LcbHLH120, LcbHLH121, LcbHLH122, LcbHLH123, LcbHLH124, and LcbHLH125. LcbHLH122 and 123 had no loop region and two helix regions. LcbHLH120, LcbHLH121, LcbHLH124, and LcbHLH159 had no loop region and lacked the second helix region. Multiple sequence alignment showed that 24 amino acid residues were highly conserved (>60% consensus ratios), and two were conserved with a 100% consensus ratio. It is noteworthy that most LcbHLH proteins have the highly conserved H–ER–RR structures and Leu-27 in Helix 1 and Leu-59 in Helix 2 of the HLH region, which are considered to play an essential role in protein dimerization. Thus, we speculated that the LcbHLH proteins may be able to form protein complexes.

The conserved motifs, domain distribution, and gene structure of *L. chinense* var. *rubrum* were analyzed to obtain more information about the bHLHs. As shown in Appendix A, 10 motifs were identified, marked as 1–10 (Table 2). The different motifs correspond to their bHLH superfamily domains. For example, we can clearly see that Motif 8 represents the bHLH–MYC_N domain, and Motif 1 and Motif 2 can form various structural domains of the bHLH. In combination with protein sequence alignment, most of the LcbHLH proteins combined Motifs 1 and 2 due to the high conservation of the basic region and the two helix regions, while LcbHLH122, LcbHLH123, LcbHLH129, LcbHLH128, LcbHLH131, LcbHLH81, LcbHLH76, LcbHLH159, LcbHLH55, LcbHLH54, LcbHLH108, LcbHLH148, LcbHLH146, and LcbHLH93 had only Motif 1. The LcbHLH proteins clustered in the same subfamily often have similar motifs. For example, Motifs 1, 2, 5, 4, 9, 6, and 10 were identified in Subfamily II, and Motifs 1, 2, and 8 were identified in Subfamily III(d + e + f).

The evolutionary relationship between gene family members can be revealed through the analysis of gene structure. The gene structure of LcbHLHs in the same subfamily was always consistent. The exon number of LcbHLHs varied from 1 to 13, whereas the exon–intron organizations were evolutionarily related (Appendix A). The LcbHLHs with one exon were clustered in four subfamilies (III(d + e), VIIIb, orphans, and II) and all the members of Subfamily VIIIb and the orphans had only one exon.

### 2.3. Analysis of Gene Duplication and Collinear Correlations

Tandem and fragment replication are critical means of amplifying a gene family. The 165 *LcbHLHs* in our study were unequally distributed among the 12 *L. chinense* var. *rubrum* chromosomes, with a maximum of 22 on chr3 and a minimum of 2 on chr4. Tandem replication is thought to occur when the distance between genes is less than 100 kb, and seven pairs of *LcbHLHs* fell into that category (Appendix A). An intraspecific collinearity study revealed that fragment replication produced 56 pairs of *LcbHLHs* (Appendix A). The findings showed that tandem replication and fragment replication were critical processes for expanding the *LcbHLH* gene family. In addition, the substitution rate ratio, Ka/Ks, is a criterion applicable to the selective pressure of gene duplication [38]. Ka/Ks values less than 1 often represent negative selection, those equal to 1 represent neutral selection, and those greater than 1 represent positive selection. Among the 56 gene pairs from fragment replication and the seven gene pairs from tandem replication, all except the pairs of *LcbHLH29/30*, *LcbHLH140/141*, *LcbHLH142/144*, and *LcbHLH102/103* among the fragment replication pairs were more significant than 1; the other Ka/Ks values were less than 1. It was speculated that they might have been purified and selected during evolution, indicating that most bHLH genes evolved slowly (Appendix A).

To better understand the genetic differentiation, gene replication, and evolution of the bHLH gene families of *L. chinense* var. *rubrum*, *Arabidopsis*, *Oryza sativa*, *Zea mays*, and *Vitis vinifera*, MCScanX (Ver 2.0) was used to analyze the homologous *bHLH* genes of these species. In total, 89, 44, 41, and 123 pairs of orthologous *bHLH* genes were detected in three comparisons (*L. chinense* var. *rubrum* vs. *Arabidopsis*, *L. chinense* var. *rubrum* vs. *Oryza sativa*, *L. chinense* var. *rubrum* vs. *Zea mays*, and *L. chinense* var. *rubrum* vs. *Vitis vinifera*, respectively) (Figure 4, Appendix A). Therefore, the *bHLH* genes of *L. chinense* var. *rubrum* and *Vitis vinifera* were considered to be more closely related than the *bHLH* genes of *L. chinense* var. *rubrum* and those of *Oryza sativa*, *Zea mays*, or *Arabidopsis*.

### 2.4. GO Annotation and Analyses of the Cis-Regulatory Elements

Highly divergent sequences outside the conserved structural domain of bHLH suggested that the LcbHLH protein may be involved in multiple biological processes to some extent. To clarify the particular functions of the bHLH proteins, a GO annotation of the LcbHLHs was performed. As shown in Appendix A, most LcbHLH proteins were annotated as being associated with transcription factors and the activity of protein dimerization, the regulation of transcription, and developmental processes. Regarding the cellular components, 139 of the 158 bHLH genes from the transcriptomic data were located in the nucleus. Only a tiny number of genes were predicted to be distributed in the organelles, such as cytosol (two), chloroplast envelope (one), nucleoplasm (one), vacuole (one), and intracellular membrane-bounded organelle (one). In addition, a few LcbHLHs were also located in less common membrane structures and complexes, including the integral component of the membrane (one), the membrane (one), and the vacuolar membrane (one). It was expected that most of the LcbHLH proteins would be engaged in protein dimerization activity (152/158) according to the molecular functions, and the LcbHLHs also participated in various biological processes. Many LcbHLH proteins were annotated as being associated with DNA binding, growth regulation, and responses to stimuli. For example, 127 LcbHLHs were associated with DNA-binding TFs; some of the LcbHLH proteins were involved in growth and development, including the regulation of change (six), pollen development (ten), development of the anther wall tapetum (eleven), and negative regulation of seed germination (six). Several LcbHLHs could react to environmental stimuli such as cold (seven), iron (five), and light (two).

The function of gene family members is often associated with cis elements. We looked for cis elements in the promoter regions of the various *LcbHLH* genes to see whether there were any distinctions in their gene regulation. As shown in Appendix A, light response elements were the most common cis elements in the *LcbHLH* promoter sequences, including Box 4, the Gap-box, the G-Box, the AE-box, the I-box, the GT1 motif, and the TCCC motif. In addition, the cis elements that participated in growth and development and related to different stress responses were also discovered, such as MSA-like, which is involved in cell cycle regulation; Motif I and the RY-element, which is engaged in seed-specific regulation; the GCN4 motif, which is involved in development of the endosperm, etc. Moreover, various cis elements were related to different stress responses, including the abiotic response elements involved in the responses to abscisic acid (ABRE), auxin (AuxRR-core, TGA-element, TGA-box, and AuxRE), jasmonic acid (CGTCA motif and TGACG motif), and gibberellin (GARE motif, TATC-box, and P-box), and the biological response elements related to salt stress (DRE) and low temperature (LTR). We also identified some cis elements with MYB recognition sites, which can regulate the synthesis of plant flavonoids (MBSI) by binding MYB-related genes.

### 2.5. Expression Profiles of LcbHLHs and qRT–PCR Analysis of Gene Expression

The expression patterns of *LcbHLHs* were analyzed on the basis of the transcriptome data of different *Loropetalum chinense* cultivars. New varieties of *Loropetalum chinense* include “Xiangnong Nichang” (XNNC), “Xiangnong Fengjiao” (XNFJ), “Xiangnong Xiangyun” (XNXY), and “Xiangnong Xiaojiao” (XNXJ) (Appendix A, Figure 5A). To better observe the differences in the expression of *LcbHLHs*, we standardized the FPKM value and removed the *LcbHLHs* with FPKM values less than one from all samples, as this value is usually considered an invalid expression. An expression pattern map of 105 LcbHLHs was established from those that remained.

It was found that the expression levels of *LcbHLHs* in flowers, young leaves (Stage I), and mature leaves (Stage II) displayed significant differences, as shown in Figure 5B. Five groups of *LcbHLHs* were divided according to the cluster analysis. Genes in Groups IV and V were highly expressed in the flowers and less represented in the leaves, and they were mainly predicted to involve the formation and development of floral organs. For example, *LcbHLH133/145/25/102/69/43* in Group V was upregulated in the flowers of XNXY and XNFJ, while the corresponding leaves showed lower expression levels. *LcbHLH84* in Group IV was highly expressed in the flowers of all varieties, *LcbHLH43* was significantly expressed in the flowers of XNFJ, and *LcbHLH22/51* was highly expressed in the flowers of XNXY and less expressed in its leaves. According to the GO enrichment analysis, *LcbHLH84* may be involved in photoperiod and flowering processes, while *LcbHLH69* can regulate pollen development. In addition, *LcbHLH51/25/102/43* can participate in gibberellin-related anabolism and mediate the brassinosteroid signaling pathway. Analysis of the cis elements showed that all of these genes had light-responsive elements and some hormone-response elements, such as those related to gibberellin, jasmonic acid, salicylic acid, auxin, and abscisic acid. Previous studies have shown that plant flowering is regulated by various plant hormones. Among them, gibberellin plays an essential role in flowering, and other hormones such as abscisic acid, auxin, salicylic acid, and jasmonic acid are also involved in regulating flowering. These results were somewhat consistent with the GO enrichment and transcriptome analyses.

Since the ornamental value of *L. chinense* var. *rubrum* mainly lies in the leaf color, three genes (*LcbHLH4/156/157*) from Subfamily IIIf were selected as being involved in anthocyanin and flavonoid biosynthesis according to the structural and functional predictions. We preliminary analyzed the gene expression profiles of varieties with different leaf color in Stages I and II (Figure 5B). *LcbHLH4* was more highly expressed in purple leaves (XNNC) than in green leaves (XNXY) and was significantly downregulated as the leaf color became green from Stage I to Stage II. However, *LcbHLH156* and *LcbHLH157* were highly expressed in Stage I of different dark-leafed varieties (XNFJ, XNXJ, and XNNC) and decreased in expression as the leaves became green during Stage II. In addition, it is also worth noting that *LcbHLH149/114* was also highly expressed in XNFJ, and *LcbHLH117* also showed high expression levels in dark-leafed varieties (XNNC). These expression levels of these genes showed similar trends to the phenotype of leaf color changes. Thus, we further studied them as candidate genes.

To further validate the relationship between *LcbHLHs* and leaf color in *L. chinense* var. *rubrum*, we demonstrated the expression levels of the candidate genes of XNFJ from the transcriptomic data using the qRT–PCR technique. These genes had consistent expression trends at the two developmental stages (I and II) (Figure 6, Appendix A). We could clearly see that the expression of these six candidate genes was higher in the young leaf stage (I) and significantly decreased in the leaf maturation stage (II), consistent with the phenotypic trend. In addition, we compared the gene expression levels of three different colored leaves of HYJM1 under natural conditions, with green leaves (GL) as the control group (Figure 7A). The results showed that these six genes were upregulated in mixed leaves (ML), especially *LcbHLH4/156/157*, which was most significantly upregulated in mixed leaves and purple leaves (ML and PL). Notably, the expression levels of *LcbHLH114/147/149* were slightly upregulated, but not significantly in mixed leaves (ML), and their expression was significantly reduced in purple leaves (PL). This suggests that they are not directly involved in the positive regulation of anthocyanin synthesis but may act in other ways to affect the anthocyanin synthesis pathway. These assumptions need further analysis and verification. (Figure 7B, Appendix A).

### 2.6. The Protein–Protein Interaction Network of the Candidate Genes

To further clarify the function, we constructed a protein interaction network of these six candidate genes via the *Arabidopsis* protein database. As shown in Figure 8, LcbHLH4 (TT8 in *Arabidopsis*) can coordinate with TTG1 and TT2 to ensure that DIHYDROFLAVONOL–4–REDUCTASE (DFR) is correctly expressed and then regulate the flavonoid pathway. LcbHLH114 (AT1G01260/bHLH13 in *Arabidopsis*) plays an important role in plants’ growth and defense. It can interact with JAZ proteins and mediate the JA response to regulate plant growth and defense [39]. In addition, it is worth noting that LcbHLH117/149 (MYC2 in *Arabidopsis*) can also interact with JAZ proteins [40,41,42], and the function of LcbHLH156/157 (GL3 in *Arabidopsis*) may be related to the regulation of trichome development [43,44]. The most interesting thing is that there is still a particular relationship among these six candidate proteins involved in different pathways, which needs further analysis and verification.

### 2.7. Overexpression of LcbHLH156

Based on the expression profile and protein interaction analysis, *LcbHLH156* from the IIIf subfamily with high expression was selected for transient expression verification. The leaves of the variety “Xiangnong Fendai” (XNFD) were overexpressed by Agrobacterium injection, and the control CK was mainly injected with water. It can be observed that compared with CK, vector control (pCAMBIA1305–GFP), and an overexpression of *LcbHLH156* (pCAMBIA1305–*LcbHLH156*) showed red accumulation at the wound, and the accumulation of overexpression plants was more evident (Figure 9A). One week after injection, the wound site of the leaves was collected by a puncher to determine the total anthocyanin content. It was found that the anthocyanin content in the leaves of overexpressing plants showed a significant increase (Figure 9B, Appendix A). At the same time, compared with CK, it was found that the overexpressed leaves showed deeper red (a*), higher yellow (b*), and similar brightness (L*) around the wound (Figure 9C, Appendix A). To further verify the results, we used qRT–PCR to detect the infected plant leaves. The results showed that compared with CK and vector control, the expression of the *LcbHLH156* gene in overexpression plants increased significantly and reached the highest expression level on the 4th day (T4) (Figure 9D, Appendix A). The above results indicate that *LcbHLH156* can promote anthocyanin synthesis.

## 3. Discussion

### 3.1. Systematic and Comprehensive Genome-Wide Detection of LcbHLHs in L. chinense var. rubrum

With the completion of the whole genome sequences of a large number of species, a large number of studies have been conducted on the *bHLH* family of various species, such as *Arabidopsis* (*n* = 162) [8,9], rice (*n* = 167) [10], bamboo (*n* = 448) [34], grape (*n* = 94) [35], and maize (*n* = 208) [45]. (The “*n*” here refers to the number of *bHLH* genes.) Compared with the number of these species, the number of *LcbHLHs* is appropriate, but the quantity is much smaller than that of bamboo and maize. Changes in the number of *bHLH* genes between species were often thought to be possibly associated with gene duplication events or genome size [46]. Our study identified 165 *bHLH* genes of *L. chinense* var. *rubrum*. According to our evolutionary analysis, these 165 *bHLHs* are divided into 21 subfamilies and the number and distribution of these subfamilies are similar to those in *Arabidopsis*. They all have more members in VII(a + b) and XII, III(a + b) subfamilies. It is worth noting that *LcbHLHs* lack XIII, XIV, and XV subfamilies, but it has been a significant increase in the number of subfamilies III(a + b + c), IVa, and XII. The obtained genes offset the loss of genes to a certain extent so that the overall number of genomes remains stable [47]. Protein multisequence alignment also confirmed that LcbHLHs had a typical bHLH domain. Further structural analysis revealed that most LcbHLHs were linked to DNA-binding and homodimer formation functions. Additionally, an analysis of the conserved motifs discovered that most of the LcbHLH family members had the associated conserved motifs (Motif 1 and Motif 2), constituting the structural domain of bHLH [5]. The collinearity analysis showed that there were 89, 44, 41, and 123 *bHLH* collinear pairs between *L. chinense* var. *rubrum* and *Arabidopsis thaliana*, *L. chinense* var. *rubrum* and *Oryza sativa*, *L. chinense* var. *rubrum* and *Zea mays*, *L. chinense* var. *rubrum,* and *Vitis vinifera*, respectively. The evolutionary distance between species is closely related to the collinearity between plant genomes. Generally speaking, the larger the evolutionary distance is, the fewer the collinearity gene pairs identified between species are [47,48]. The evolutionary distance between *L. chinense* var. *rubrum* and *Vitis vinifera* is shorter, and the genetic relationship is closer, which is consistent with the fact that they belong to woody plants and dicotyledons. These results further confirmed the accuracy of our analysis of the *bHLH* gene family in *L. chinense* var. *rubrum* provided a basis for further applications.

### 3.2. Functional Prediction of LcbHLHs

As plants’ second-largest gene family, *bHLHs* always play an irreplaceable role in plants’ growth, development, and survival [5,11]. In our study, GO annotation and cis-element analysis predicted that *LcbHLHs* are widely involved in the growth and development of plants, including the regulation of secondary metabolite synthesis, the regulation of the cell cycle, and the response to various stresses. In addition, a few *LcbHLHs* also have some cis elements that respond to hormones, such as gibberellin, jasmonic acid, auxin, etc. To further explore the accuracy of our functional analysis, we analyzed the transcriptomic data and validated some of them by using real-time quantitative PCR.

Many *LcbHLHs* were expressed differently in different tissues. Some *LcbHLHs* were highly expressed in the flowers of the three varieties. *LcbHLH106* was annotated as PIF3, a phytochrome-interacting factor necessary for photo-induced signal transduction that can further regulate flowering time [49,50]. *LcbHLH13* was highly expressed in the leaves and was annotated as PIL5, which can indirectly regulate the content of DELLA protein and further participate in the gibberellin pathway to regulate leaf elongation and other processes of growth and development (Appendix A) [51,52,53]. Moreover, the expression patterns of *LcbHLHs* in different stages and varieties were analyzed. Three genes (*LcbHLH4/156/LcbHLH157*) selected by evolutionary analysis and GO annotation showed high expression patterns in dark leaves, and *LcbHLH114/117/149* were also upregulated in dark-leafed varieties. In addition, the qRT–PCR results of six genes showed the same trend in XNFJ. They all showed higher expressions in Stage I than in Stage II, which matched the varieties’ transcriptomic results and leaf color characteristics. From the qPCR results of HYJM1 leaves with different colors, we also found that the expression trends of *LcbHLH4/156/157* genes in HYJM1 leave with different colors were generally consistent, and these genes were significantly upregulated in mixed and purple leaves.

To further demonstrate our hypothesis, we analyzed the interactions of these six bHLH proteins through a protein interaction network. It was found that these proteins were mainly involved in two pathways. LcbHLH117/149/114 were mainly involved in JA signal transduction. Previous studies have shown the bHLHs in Subfamily III(d + e) participate in the JA signal pathway, leading to the accumulation of anthocyanin in apples [54,55] and the regulation of plant defense in *Arabidopsis* [56,57,58]. In *Arabidopsis*, JAZ proteins can interact with bHLH TFs (GL3) to mediate the accumulation of anthocyanin [59,60]. In addition, previous studies have shown that members of Subfamily IIIf in the bHLH family are related to anthocyanin synthesis and trichome initiation [19,61]. AtbHLH042/TT8 could coordinate with TTG1 and TT2 to make DIHYDROFLAVO–NOL 4–REDUCTASE (DFR) and BANYULS (BAN) correctly expressed and then regulated the flavonoid pathway [23,62,63]. AtbHLH001/GL3 could interact with TTG1 to regulate trichome development [64,65,66]. It was identified as an essential regulator of the anthocyanin pathway in *Arabidopsis*, together with TTG1 and MYB75 transcription factors to regulate specific genes in the anthocyanin synthesis pathway [43,44]. In our study, *LcbHLH4/156/157* from the IIIf subfamily plays an irreplaceable role in the synthesis of anthocyanins and the development of trichomes, and it was predicted to interact with TT2/TTG1/MYB75 [21,67].

Based on the above studies, *LcbHLH156* of Subfamily IIIf was selected for verification. Through the transient expression of *L. chinense* var. *rubrum* leaves, compared with CK, the total anthocyanin content of the leaves in the transient overexpression plants increased, and there was also a significant red accumulation on the wound (a* > 0). The expression of *LcbHLH156* by qRT–PCR found that the expression level reached the highest on the 4th day (T4) after injection and decreased with time, which may be due to the transient expression. These results indicate that *LcbHLH156* is closely related to anthocyanin synthesis. In addition, these findings also provide an experimental basis for further verification of the interaction network of *LcbHLH156* regulating anthocyanin synthesis and make a reference for further exploring the mechanism of MBW complex regulating anthocyanin synthesis in *L. chinense* var. *rubrum*.

## 4. Materials and Methods

### 4.1. Plant Materials and Data Resources

Tissue specimens of *L. chinense* var. *rubrum* were obtained from the Floral Experimental Station of the College of Horticulture, Hunan Agricultural University, Changsha, China. These experimental materials included the flowers and leaves of some new *Loropetalum chinense* cultivars, namely “Xiangnong Xiangyun” (XNXY), “Xiangnong Fengjiao” (XNFJ), “Xiangnong Nichang” (XNNC), “Xiangnong Xiaojiao” (XNXJ), “Xiangnong Fendai (XNFD),” and “The No. 1 *Loropetalum chinense*” (HYJM1). The leaf material of HYJM1 mainly has three different colors in the natural state, while the leaf material of other species was mainly collected during the following developmental stages: the new leaf stage (Stage I), when the leaves were soft and brightly colored, and the mature period (Stage II), when the leaves were leathery and darkly colored. Samples were obtained as three biological replicates from the four new varieties and immediately frozen in liquid nitrogen for transcriptome sequencing. The preparation of the RNA samples and library construction were consistent with previous studies [68,69].

The raw transcriptomic data involved in this article were uploaded to the Genome Sequence Archive (Genomics, Proteomics, and Bioinformatics 2021) at the National Genomics Data Center (Nucleic Acids Research 2022) and the Chinese National Center for Biological Information at the Institute of Genomics (GSA: CRA009284 and CRA009285). They can be obtained from Genome Sequence Archive of China National Center for Bioinformation (https://ngdc.cncb.ac.cn/gsa) (accessed on 10 May 2023).

### 4.2. Identification and Physicochemical Characterization of the bHLH Gene Family

The genomic data and gene annotation information of *L. chinense* var. *rubrum* were obtained from the research group of Hunan Agricultural University [70]. First, we downloaded the seed profile of the bHLH signature domain (PF00010) from the PFAM database (https://pfam.xfam.org/ (accessed on 16 April 2022)) [71] and made use of a hidden Markov model HMMER (Ver 3.0) to screen the candidate bHLH proteins with the E-value cut-off set to 10–5 [72]. Then, the *Arabidopsis* bHLH protein homology BLAST method was used to search extensively for candidate bHLH proteins (https://blast.ncbi.nlm.nih.gov/Blast.cgi (accessed on 16 April 2022)). The protein sequences of *Arabidopsis* were downloaded from TAIR (https://www.arabidopsis.org/ (accessed on 16 April 2022)), and proteins containing the bHLH domain were screened further in the databases of PFAM (http://pfam-legacy.xfam.org/ (accessed on 16 April 2022)) and SMART (https://smart.embl.de/ (accessed on 16 April 2022)). The candidate LcbHLH proteins were obtained by combining two methods (Figure 10). The subcellular localization of LcbHLHs was predicted by the online website Wolf-psort (https://wolfpsort.hgc.jp/ (accessed on 17 April 2022)). The sequences of LcbHLHs were analysed bioinformatically, and the proteins’ physicochemical parameters were calculated using ExPASy (https://web.expasy.org/compute_pi/ (accessed on 19 April 2022)) [73].

### 4.3. Chromosomal Localization, Tandem Duplication, and Collinearity of the bHLH Genes

The calculation of the chromosome distribution was performed on the GSDS website (http://gsds.gao-lab.org/ (accessed on 6 March 2023)). Gene duplication analysis of *L. chinense* var. *rubrum* was conducted with the Multiple Collinearity Scan Toolkit (MCScanX) (Ver 2.0). The tools of TBtools (Ver 2.003) and MCScanX (Ver 2.0) were used to perform the interspecies collinearity analysis of *bHLHs* between *L. chinense* var. *rubrum* and *Arabidopsis*, *Oryza sativa*, *Zea mays*, and *Vitis vinifera* [74,75]. All of these genomic data were obtained from EnsemblPlants (https://plants.ensembl.org/ (accessed on 22 November 2022)). The non-synonymous replacement rate (Ka) and synonymous replacement rate (Ks) of the replicated gene pairs were calculated using KaKs Calculator 2.0 (Ver 2.0), and the environmental selection pressure was analyzed using the Ka/Ks ratio.

### 4.4. Evolution and Multiple Sequence Alignment of LcbHLHs

DNAMAN software (Ver 6.0) was used for a multiple sequence alignment of the proteins, and SnapGene software(Ver 4.3.6) was used to plot the amino acid site distribution of the bHLH proteins’ conserved domains. The evolutionary bHLH proteins trees of *Arabidopsis thaliana, Oryza sativa,* and *L. chinense* var. *rubrum* were constructed by using MEGA11 software (Ver 11.0.11)

### 4.5. Analysis of the Gene Structure, Conserved Motifs, and Family Structural Domains

The intron/exon gene structure maps were obtained as GFF3 files via the Gene Structure Display Server (GSDS) website (http://gsds.gao-lab.org/ (accessed on 1 June 2022)). The analysis of conserved motifs was conducted via the MEME website (https://meme-suite.org/meme/doc/meme.html (accessed on 18 April 2022)). The number of motifs was 10; all other parameters were set to the default. The final result graph was drawn by using TBtools software (Ver 2.003).

### 4.6. GO Annotation, Analysis of the Cis-Acting Components, and the Protein–Protein Interactions of LcbHLHs

The NCBI database (https://www.ncbi.nlm.nih.gov/ (accessed on 16 March 2023)) was selected as a reference database for further GO analysis of *LcbHLHs* using the Blast2GO program (Ver 6.0) [76]. A 2000 bp promoter upstream of the *bHLH* gene family member of *L. chinense* var. *rubrum* was extracted by using TBtools software (Ver 2.003) and uploaded to the PlantCARE website (https://bioinformatics.psb.ugent.be/webtools/plantcare/html/ (accessed on 12 June 2022)) for predicting the cis-elements. To better understand the regulation of LcbHLHs in protein–protein interaction networks, the STRING database (STRING: functional protein association networks (string-db.org) (accessed on 15 August 2022)) was used to predict the protein interaction network of the candidate *bHLH* gene family in *L. chinense* var. *rubrum*.

### 4.7. Gene Expression Patterns and Quantitative Real-Time PCR

All raw transcriptomic data were presented as FPKM values (fragments per kilobase of transcript per million mapped reads). We homogenized the raw FPKM values and visualized the heatmap using the R package [77,78].

RNA from the plant material was extracted using the FastPure Universal Plant Total RNA Isolation Kit (Vazyme, Nanjing, China). The first standard cDNA was synthesized by using the Evo M–ML RT Kit for qPCR (Accurate Biology, Changsha, China) and stored in a freezer at −40 °C. The primers were designed using Beacon Designer 8. (Appendix A), and qRT–PCR was performed on a Bio–Rad CFX384TM (Bio-Rad, Hercules, CA, USA) with 2X SYBR Green Pro Taq HS Premix (Accurate Biology, Changsha, China). The cDNA was diluted to 500 ng, and the system was set to 10 μL to be prepared as three technical replicates. Each reaction had a 1 μL template. The conditions for qRT–PCR were as follows: 95 °C for 5 min, followed by 40 cycles of 95 °C for 15 s, 60 °C for 1 min, and 72 °C for 5 min. Quantitative PCR expression levels were calculated using the 2^−∆∆CT^ method, and the expression values of three replicates were normalized using *LcActin* as the internal control. The error bars represent the standard errors from three biological replicates.

### 4.8. Transient Expression of L. chinense var. rubrum and Color Measurement

The CDS sequence of *LcbHLH156* was amplified by Phanta Max Super–Fidelity DNA Polymerase kit (Vazyme, Nanjing, China), and the specific primers were designed by Primer5 software (Ver 5.0) (Appendix A). The CDS sequence of *LcbHLH156* was ligated to the BamHI site of the pCAMBIA1305–GFP vector by homologous recombination, and the recombinant plasmid was transformed into the GV3101 strain. The bacterial liquid was cultured in a shaker at 28 °C and 200 rpm for 24 h. When the bacterial solution was shaken to golden yellow and the OD600 value reached 1.0, the cells were centrifuged at 5000 rpm for 10 minutes. The Agrobacterium cells were collected and re-suspended using the infection solution (10 mM MgCl_2_, 10 mM MES, pH 5.8, and 200 μM acetosyringone) [79]. The leaves of *L. chinense* var. *rubrum* were injected with a 1 mL sterile syringe, and 5–10 branches were injected into each plant. Photos were recorded every three days by a portable fluorescent protein excitation light source LUYOR–3415 (LUYOR, New York, NY, USA). Random sampling was performed at 1, 4, and 7 days after treatment for qRT–PCR expression detection.

To analyze the phenotypic data, we used a puncher to sample the wound site on the leaves one week after injection. Leaf samples (10 mg) were extracted with 1 mL of 0.5% HCl in methanol for 1 day at 4 °C in the dark. After centrifuging at 12,000 rpm for 20 min, the absorbance at 510 nm and 700 nm was measured in pH 1.0 and 4.5 buffers by UV–2600 Spectrophotometer. Each sample was repeated three times [80].

### 4.9. Statistical Analysis

All the experiments included three replicates, and the results were presented as mean ± SD of three independent experiments. Student’s *t*-test statistically analyzed the data. An asterisk (*) indicates *p* < 0.05, a double asterisk (**) indicates *p* < 0.01, and a triple asterisk (***) indicates *p* < 0.001.

## 5. Conclusions

In summary, we analyzed the physicochemical properties, evolution, gene structure, Ka/Ks value, and collinearity of the bHLH family members of different *L. chinense* var. *rubrum* varieties. In addition, we also predicted the function of LcbHLHs by using cis elements, evolutionary analysis, and GO annotation. Six candidate genes were selected by analyzing the evolutionary tree and the expression patterns of 105 *LcbHLHs* in different tissues and varieties. Through further analyses of qPCR and predictions of the protein interactions, we obtained a preliminary understanding of the function of these six candidate genes. The *LcbHLH156* candidate gene of the IIIf family was selected for transient expression to verify the function. It was found to promote anthocyanin synthesis through phenotypic analysis and expression analysis. All these results strengthen our understanding of the bHLH gene family in *L. chinense* var. *rubrum* and provide a basis for further studies on the mechanism regulating the change in leaf color in *L. chinense* var. *rubrum*.

## Figures and Tables

**Figure 1 plants-12-03392-f001:**
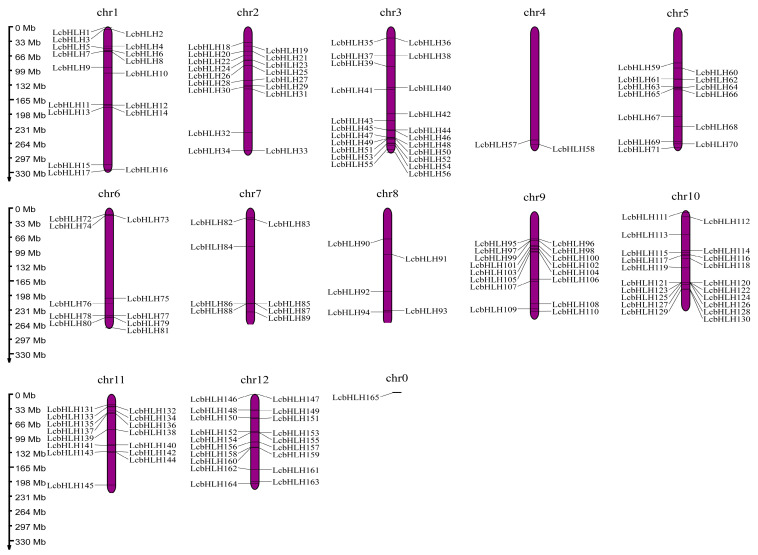
Chromosomal distribution of LcbHLHs. The left-hand scale is used to assess the chromosome size. All chromosomes contain the *LcbHLH* genes. The scaffold is represented by chr0.

**Figure 2 plants-12-03392-f002:**
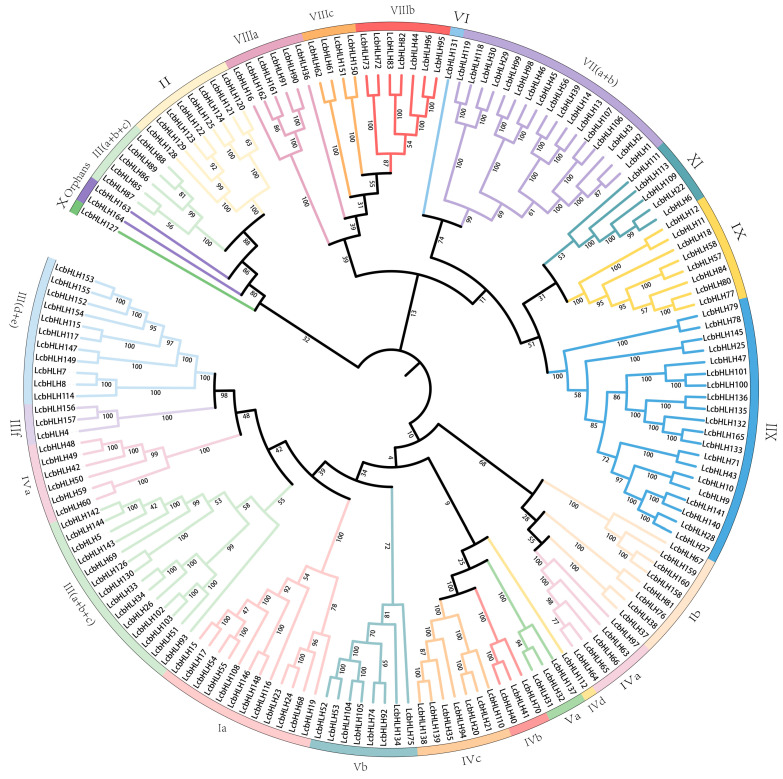
Evolutionary bHLH proteins tree in *L. chinense* var. *rubrum*. The Evolutionary tree of *L. chinense* var. *rubrum* was constructed using MEGA11 software (Ver 11.0.11) via the neighbor-joining (NJ) method. The parameters were set as follows: method, “p-distance”; treatment of gaps in the data, partial deletion; self-expansion value, “bootstrap = 1000”. Different subfamilies are represented by branches and frames of different colors.

**Figure 3 plants-12-03392-f003:**
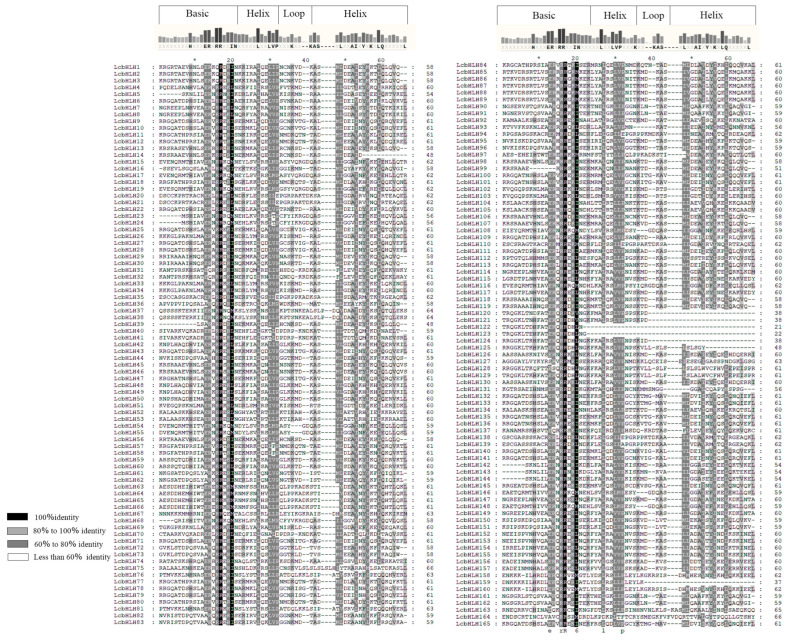
Multiple alignment of the bHLH domains in LcbHLH family proteins in the LcbHLH domain shows conserved amino acids. Sequence identities > 60% are shown in grey or black shading. ”*” was used to interval 10 bases.

**Figure 4 plants-12-03392-f004:**
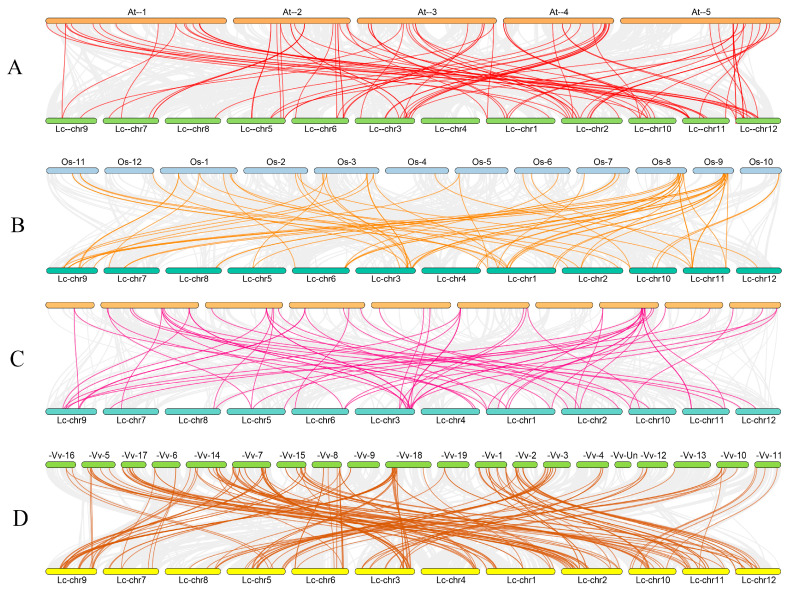
Syntenic analyses of the *bHLH* genes of *L. chinense* var. *rubrum*, *Arabidopsis* (**A**), *Oryza sativa* (**B**), *Zea mays* (**C**), and *Vitis vinifera* (**D**).

**Figure 5 plants-12-03392-f005:**
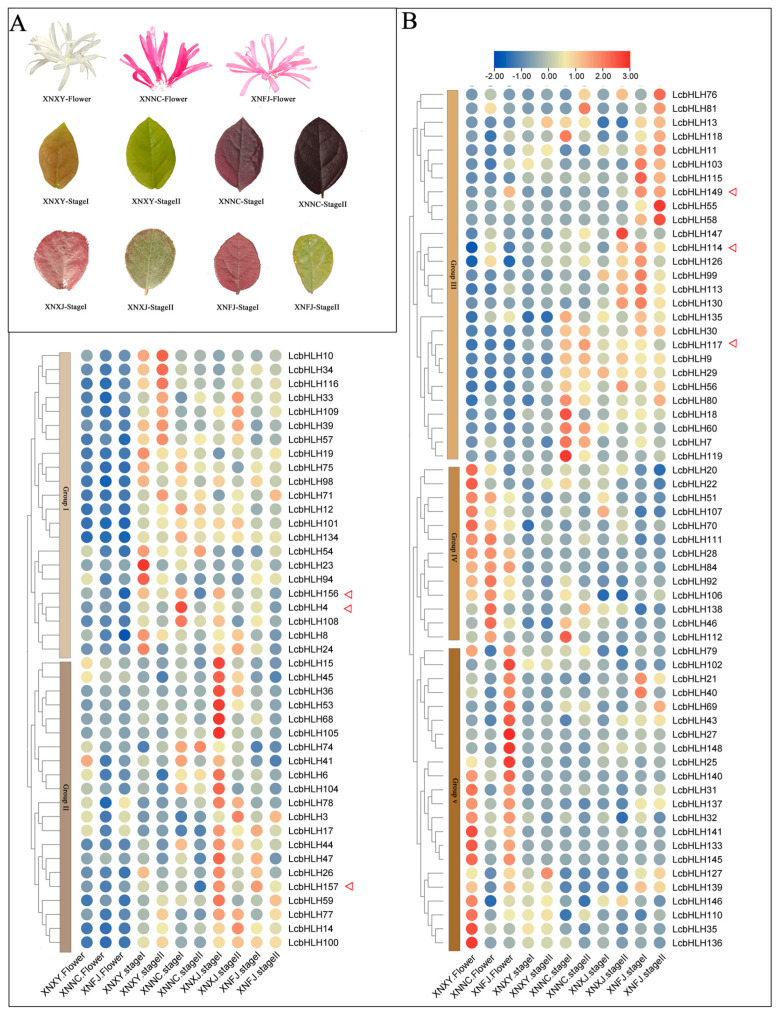
Expression patterns of *LcbHLHs* (**A**) Four varieties with different colored leaves and flowers. Stage I and II represent the first−stage and second−stage leaves, respectively. The first stage is the young leaf stage; the second is the mature leaf period. (**B**) Expression patterns of the *LcbHLHs*. Expression profiles of *LcbHLHs* in different tissues. “XNFJ,” “XNXY,” “XNXJ,” and “XNNC” are the names of our selected materials. The red triangle is used to point out candidate genes.

**Figure 6 plants-12-03392-f006:**
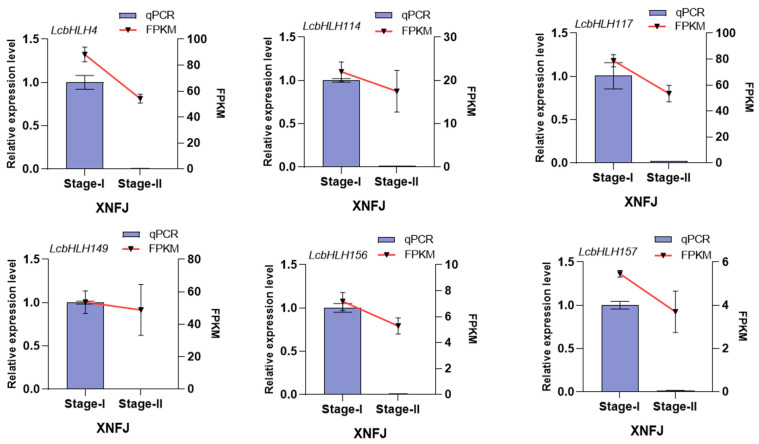
Expression of six differentially expressed *LcbHLHs* in the variety XNFJ during Stages I and II.

**Figure 7 plants-12-03392-f007:**
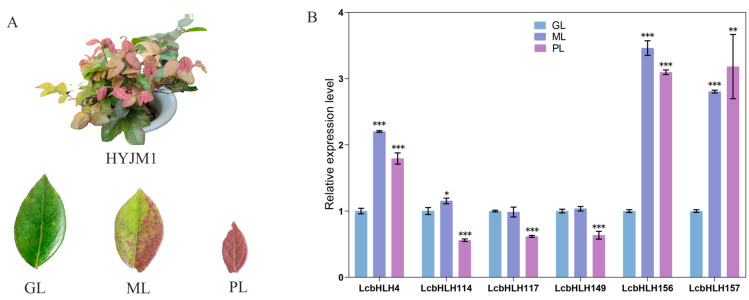
Expression of six differentially expressed *LcbHLHs* in different-colored leaves (**A**) HYJM1 varieties with different colored leaves. GL, green leaves; ML, mixed green and purple leaves; PL, purple leaves. (**B**) Expression of six *LcbHLHs* in different colored leaves (GL, ML, PL), with green leaves (GL) as the control group. An asterisk (*) indicates *p* < 0.05, a double asterisk (**) indicates *p* < 0.01, and a triple asterisk (***) indicates *p* < 0.001.

**Figure 8 plants-12-03392-f008:**
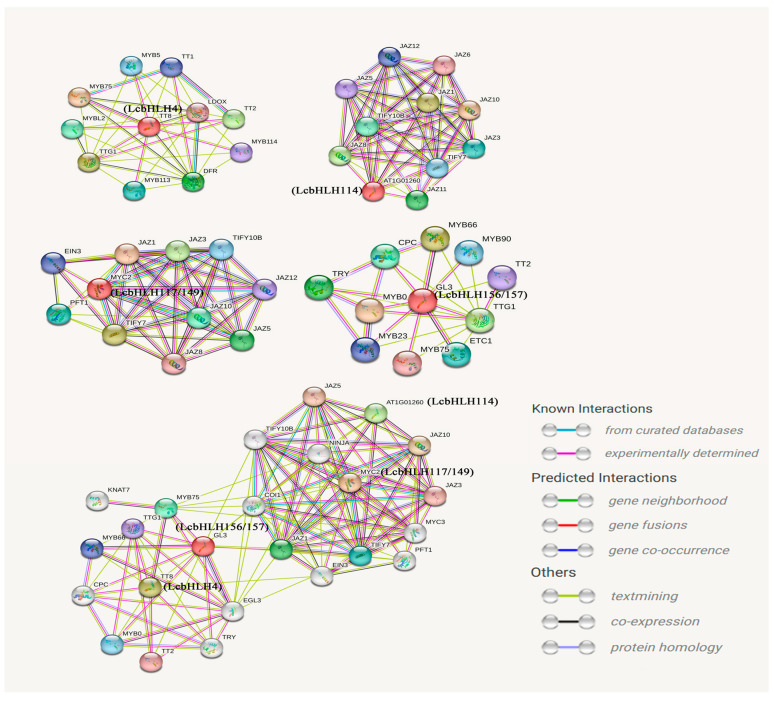
Interaction network for the six candidate LcbHLH proteins. The predicted results in parentheses are based on homologous genes in *Arabidopsis*. The colors of the lines represent the status of research into these protein interactions.

**Figure 9 plants-12-03392-f009:**
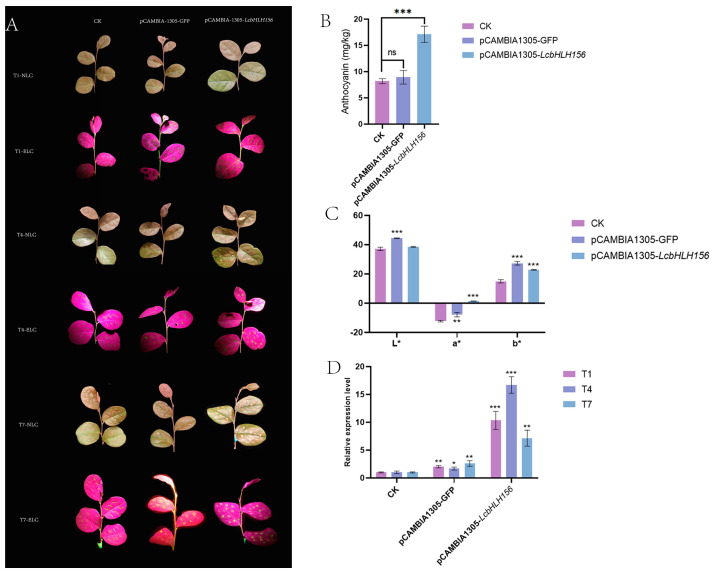
Transient expression result of *L. chinense* var. *rubrum* (**A**) The leaf phenotype map of “XNFD” within 1 week of leaf injection; “NLC” stands for natural light condition and “ELC” represents excitation light condition. (**B**) Determination of anthocyanin content around the wound after 1 week of injection. (**C**) Color parameters around the leaf wound (L*, a*, b*). (**D**) qRT–PCR detected the transcriptional level of *LcbHLH156* in overexpressing plants. “T1,” “T4,” and “T7” represent the 1st day, 4th day and 7th day after injection. “CK” was injected with water, “pCAMBIA1305–GFP” refers to vector control, and “pCAMBIA1305–*LcbHLH156*” indicated the overexpression of plants. An asterisk (*) indicates *p* < 0.05, a double asterisk (**) indicates *p* < 0.01, and a triple asterisk (***) indicates *p* < 0.001.

**Figure 10 plants-12-03392-f010:**
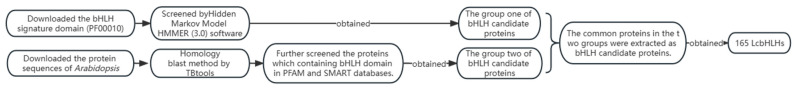
Process of identifying the bHLH family proteins.

**Table 1 plants-12-03392-t001:** Quantitative distribution of each subfamily of bHLHs in *L. chinense* var. *rubrum*, *A. thaliana*, *Ficus carica* L., and *Oryza sativa*.

Subfamilies	Number of *LcbHLHs*	Number of *AtbHLHs*	Number of *FcbHLHs*	Number of *OsbHLHs*
Ia	12	10	8	15
Ib	8	13	7	7
II	8	4	0	2
III(a + b + c)	19	10	9	10
III(d + e)	11	8	6	5
IIIf	3	4	1	7
IVa	11	4	4	6
IVb	3	3	4	4
IVc	7	4	3	4
IVd	1	2	1	9
Va	3	3	3	3
Vb	8	5	6	9
VI	1	2	2	0
VII(a + b)	17	15	7	14
VIIIa	6	4	3	1
VIII(b + c)	11	11	14	18
IX	8	6	5	5
X	1	10	14	14
XI	5	5	4	7
XII	20	17	13	18
XIII	0	3	4	3
XIV	0	3	0	3
XV	0	4	0	6
Orphans	2	9	0	3

**Table 2 plants-12-03392-t002:** Sequences of the 10 predicted motifs of the LcbHLH proteins.

Motif	Sequence
1	SHSLAERRRRERJNERFKALRSLVPNCSK
2	MDKASMLDEAIEYVKELQRQVQELSMKLE
3	EEPKSDYIHVRARRGQATD
4	QVMSFEQSNWDASVHEIQGMTSFEHPHNQDQQLHLLHEMQQNGHHHPQSF
5	FVQKPANFQTSLGFLGDLPTPDNASASSVLYDPLFHLNLPPQPPLFRDLF
6	YNLPASRTASLFGGGIDEKEGSGGVYQNGVATQFDNGVLEFTGDIGGMGK
7	RJVSALEKLGLDIVHANVSTF
8	ERAKLAKSAGIRTLVCIPTASGVVELGSTEIIKEDLGLIQLIKSLF
9	NSSTLPDTSPYINNPPTQHLLNLFHLPRCSPSSNLLPNSSI
10	DNIRLSMEELSYHQNPHQEDDAALEQHLGFDMENCYNINNN

## Data Availability

Data are contained within the article and the Appendix A.

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
