# Peer review of "Genome-Wide Analysis of the bHLH Gene Family in Loropetalum chinense var. rubrum: Identification, Classification, Evolution, and Diversity of Expression Patterns under Cultivation"

_plants, 2023, doi:10.3390/plants12193392_

Round 1
Reviewer 1 Report (Previous Reviewer 3)
Comments,
1) Describe how your research contributes to current knowledge or addresses a gap in the literature.
2) Moderate editing of English language is needed.
3) Future impacts should be clearly stated.
4) Figure 4. The figure is not clear. Not Readable.
5) The discussion section remains to have insufficient depth.
6) What is the significance of study?
Moderate editing of English language is needed.
Author Response
Dear Reviewer,
Thank you for your detailed review of our manuscript entitled “Genome-wide analysis of the bHLH gene family in Loropetalum chinense var. Rubrum: Identification, classification, evolution, and diversity of expression patterns under cultivation” (plants-2601549). The comments are of great help to improving the manuscript. We have studied the comments carefully and performed corresponding corrections in the revised manuscript. The point-by-point responses to the comments and suggestions are listed below.
We sincerely thank the editor and all reviewers for your valuable feedback that we have used to improve the quality of our manuscript. The reviewer comments are listed in standard black font and specific questions have been numbered. Our response is given in red font and the revisions of the manuscript are highlighted in blue.
Suggestions for Authors
1) Describe how your research contributes to current knowledge or addresses a gap in the literature.
2) Moderate editing of English language is needed.
3) Future impacts should be clearly stated.
4) Figure 4. The figure is not clear. Not Readable.
5) The discussion section remains to have insufficient depth.
6) What is the significance of study?
Comments on the Quality of English Language
Moderate editing of English language is needed.
Responses to reviewer (original comments by reviewer are in black color)
Q1:Describe how your research contributes to current knowledge or addresses a gap in the literature.
Reply: We sincerely appreciate the valuable comments. In the current research reports on the leaf color of Loropetalum chinense var.rubrum, most of the studies are based on physiological and phenotypic levels. For example, the color change of L.chinense var. rubrum under heat treatment[1] and different light quality treatments [2]. All of these studies are based on phenotypic analysis and physiological measurements. At the molecular level, they only involve structural genes in the anthocyanin synthesis pathway (such as LcCHI, LcANS, LcCHS, LcDFR[3,4]), and the research on the regulatory pathway is not yet clear. In our study, we filled the gap in the study of the bHLH gene family of L.chinense var. rubrum, and screened six candidate genes based on genomic and transcriptome data. The function and interaction relationship of candidate proteins were predicted by protein interaction network analysis, and the candidate gene LcbHLH156 related to anthocyanin synthesis was selected for preliminary verification. These results provide a theoretical and practical basis for further study on the regulation mechanism of leaf color in L.chinense var. rubrum. (Page 2 Line 93-101, Page 3 Line102-121).
Q2:Moderate editing of English language is needed.
Reply: Thanks for your suggestion. We apologize for the poor language of our manuscript. We are constantly revising the manuscript and polished the language through professional institution of MDPI editor service, which is well recognized and accepted by the public and has a good reputation. In the subsequent revisions, we also invited professors with native speakers (with the help of professor Donglin Zhang, a horticulturist in University of Geogia, USA; Associated professor Marek Mutwil, a bioinformatic analysis scientist in Nanyang Technological University, Singapore) to polish our manuscript. Here we did not list the changes but marked in blue in the revised paper. We appreciate for your warm work sincerely and hope that the correction will meet with approval.
Q3: Future impacts should be clearly stated.
Reply: Thanks for your kindly suggestions. We have modified and improved in the discussion section, and the further development of the experiment is envisaged. In this study, bioinformatics methods were used to identify and analyze the bHLH gene family of L.chinense var. rubrum. Through phylogenetic tree, expression profile and protein interaction analysis, six candidate genes were screened and LcbHLH156 was preliminarily verified. These results indicate that the bHLH gene family plays an important role in anthocyanin synthesis. In the follow-up study, we can deeply explore the LcbHLH156 (GL3 in Arabidopsis thaliana) protein regulatory network and verify its connection with TTG1/MYB75. In addition, we can further explore the interaction between GL3 and JAZ protein in L.chinense var. rubrum, and deeply understand the MBW complex regulating anthocyanin synthesis pathway[5–7]. These results provide reference value and experimental support for the development of leaf color regulation mechanism of L.chinense var. rubrum and other color leaf plants. (Page 16 Line 447-457, Page 17 Line 458-474)
Q4: Figure 4. The figure is not clear. Not Readable.
Reply: Thanks for your kindly suggestions. According to your suggestions, we have adjusted the Figure 4 accordingly. However, because of the large number of bHLH members, the readability of images and texts is poor. I tried to increase the size of the motif plate and chose a more contrasting color palette to display the motif and phylogenetic tree classification section for reading, but with little success. In addition, I also tried to cut the picture and display it in parallel, but the readability of the text still did not improve. In order to allow readers to see the picture, I decided to upload Figure 4 pictures as an attachment (FigureS2) to display, which is more conducive for readers to read better. Here sincerely thank you for your understanding and tolerance.
Q5: The discussion section remains to have insufficient depth.
Reply: Thank you for your suggestion, according to your suggestion, the discussion section was revised. In the first section of the discussion, the analysis of bHLH gene family was further improved, and the results of collinearity analysis were added. Based on the reading of previous literature, the distribution characteristics and evolutionary relationship of bHLH gene family members in L.chinense var. rubrum were further compared and discussed. In addition, in the second section of the discussion, we elaborated the function of the bHLH gene family in L.chinense var. rubrum, and analyzed the interaction relationship of the candidate proteins through the protein interaction network. According to the previous research results, the hypothesis of regulating the network relationship of anthocyanin synthesis pathway was proposed (it was recorded in the previous literature that GL3 plays an important role in the regulation of anthocyanin, mainly involving the formation of MBW complex and the interaction with JAZ and MYB in Arabidopsis[5,6]). In our study, LcbHLH156 is homologous to GL3 in Arabidopsis. We initially verified it and found that anthocyanin increased after transient expression of LcbHLH156. This is consistent with the results of the hypothesis, However, the regulatory pathways and interactions involved need to be further verified. These results and analysis provide experimental basis and theoretical support for the subsequent research on the regulation mechanism of leaf color of L.chinense var. rubrum. (Page 15 Line 396-404, Page16 Line 405-421, Page 16 Line 447-457, Page 17 Line 458-474)
Q6: What is the significance of study?
Reply: The particular significance of this study lies in filling the gap in the study of bHLH gene family in L.chinense var. rubrum, and promoting the study of the molecular mechanism of leaf color regulation in L.chinense var. rubrum is a highly ornamental plant, and its leaf color change is affected by a variety of structural genes and transcription factors. At present, the research on L.chinense var. rubrum is mostly phenotypic observation and physiological index determination[1,2]. At the molecular level, it is limited to the study of WRKY gene family[8] and the cloning of anthocyanin synthesis structure genes (CHS、ANS、CHI、ANS etc)[3]. Therefore, identification and analysis of the bHLH gene family of L.chinense var. rubrum has greatly promoted the progress in the molecular research field of L.chinense var. rubrum In addition, the leaf color of plants is often the focus of researchers, and the bHLH gene family has been shown to regulate anthocyanin synthesis. The analysis and verification of the function of LcbHLH156 can provide theoretical basis and experimental support for the further development of leaf color regulation mechanism in L.chinense var. rubrum and other color leaf plants.
- Cai, W.; Zhang, D.; Zhang, X.; Chen, Q.; Liu, Y.; Lin, L.; Xiang, L.; Yang, Y.; Xu, L.; Yu, X.; et al. Leaf Color Change and Photosystem Function Evaluation under Heat Treatment Revealed the Stress Resistance Variation between Loropetalum Chinense and L. Chinense Var. Rubrum. PeerJ 2023, 11, e14834, doi:10.7717/peerj.14834.
- Zhang, Y.; Liu, Y.; Ling, L.; Huo, W.; Li, Y.; Xu, L.; Xiang, L.; Yang, Y.; Xiong, X.; Zhang, D.; et al. Phenotypic, Physiological, and Molecular Response of Loropetalum Chinense Var. Rubrum under Different Light Quality Treatments Based on Leaf Color Changes. Plants 2023, 12, 2169, doi:10.3390/plants12112169.
- Li, C. Cloning and genetic transformation of LcCHSs and LcDFRs genes from Loropetalum chinense var.rubrum. Master, Hunan University of Technology, 2021.
- Zhang, X. Cloning, Expression and Transformation of Lc CHI and Lc ANS Genes from Loropetalum Chinense Var. Rubrum. Master, Hunan University of Technology, 2020.
- Wen, J.; Li, Y.; Qi, T.; Gao, H.; Liu, B.; Zhang, M.; Huang, H.; Song, S. The C-Terminal Domains of Arabidopsis GL3/EGL3/TT8 Interact with JAZ Proteins and Mediate Dimeric Interactions. Plant Signaling & Behavior 2018, 13, e1422460, doi:10.1080/15592324.2017.1422460.
- Song, S.; Liu, B.; Song, J.; Pang, S.; Song, T.; Gao, S.; Zhang, Y.; Huang, H.; Qi, T. A Molecular Framework for Signaling Crosstalk between Jasmonate and Ethylene in Anthocyanin Biosynthesis, Trichome Development, and Defenses against Insect Herbivores in Arabidopsis. Journal of Integrative Plant Biology n/a, doi:10.1111/jipb.13319.
- Gonzalez, A.; Zhao, M.; Leavitt, J.M.; Lloyd, A.M. Regulation of the Anthocyanin Biosynthetic Pathway by the TTG1/BHLH/Myb Transcriptional Complex in Arabidopsis Seedlings. The Plant Journal 2008, 53, 814–827, doi:10.1111/j.1365-313X.2007.03373.x.
- Liu, Y.; Zhang, Y.; Liu, Y.; Lin, L.; Xiong, X.; Zhang, D.; Li, S.; Yu, X.; Li, Y. Genome-Wide Identification and Characterization of WRKY Transcription Factors and Their Expression Profile in Loropetalum Chinense Var. Rubrum. Plants 2023, 12, 2131, doi:10.3390/plants12112131.
Once again, thank you very much for your comments and suggestions. A revised manuscript is attached. Should you have any questions, please contact us without any hesitation.
Sincerely yours,
Corresponding author
Yanlin Li

Reviewer 2 Report (Previous Reviewer 1)
The authors addressed all the criticisms in the revised version of the manuscript. The changes made improved the quality of the manuscript and validated the results obtained.
Author Response
Dear reviewer,
Thank you very much for your help!
Best wishes!
Yanlin Li
This manuscript is a resubmission of an earlier submission. The following is a list of the peer review reports and author responses from that submission.
Round 1
Reviewer 1 Report
Mayor comments
In order to clarify the evolutionary relationship of L. chinense var. rubrum the authors construct a phylogenetic tree using Arabidopsis, why they did not include other species? this would provide a more accurate classification of the family members
Figure 4 needs some graphical improvements, as increase size of the motif legend, which is difficult to read in the current format. I would also suggest to use a more contrasting colour palette to represent the motifs in panel B. Also, why the motifs are not organised in progressive order in the legend? The legend needs to be revised: It seems that part of the description for panel C belongs to panel D.
The section on the expression data is the most interesting part of the manuscript and needs some revisions to be fully understood by the readers. For example, in Fig7 the material used for the trancriptomics should be better illustrated, the naming of the samples is not immediately clear to the reader, I suggest to extend the description as “XNFJF" could be “XNFJ-Flower" and “XNFJI” into “XNFJ-stageI. The text in panel B is too small, it is difficult to see the genotype labels. I suggest to use a vertical heat map instead of a circular one to better display the data. “XNFJI” appears twice, which makes difficult for the reader to interpret the data and discriminate between the two leaf.
At Page 12 line 325 “The results showed that these six genes were differentially up-regulated..” . This is not consistent with the data shown in figure 8. Where 3 genes show up-regulation and 3 down-regulation. That authors should revised the text accordingly. Finally, the authors should better explain why for one genotype a down-regulation of the candidate genes is shown and in the other an up-regulation is observed.
The paragraph 2.6. The protein-protein interaction network of candidate gene. need to be revised, the interaction found are not described in sufficient details. It seems that the in network is based on studies in Arabidopsis. What is the evidence for the conserve interaction? It is more a paragraph for the discussion section than a real result paragraph. I suggest to move the observation to the discussion this will help to put the finings in a broader context.
Minor comments
Page 1 line 24: “second-largest transcription factor in plants”. Add “family”
Page 3 line 105: delete “and so on”.
Page 3 lne 131 “Oryza sativa” change to italic
Page 5 line 152. remove “Arabidopssi thaliana” from the figure to legend, in Fig2 only bHLH proteins in L. chinense are shown.
page 10 line 247, delete “transcription factor complex (1) and RNA polymerase II transcription factor complex (6).” Most bHLH are transcription factors, it is expected that re part of TF complexes.
Page 10 line 256. Remove “etc”.
Page 10 line 261. Remove “and so on”.
Page 11 line 268. Change “MeJa” into “Jasmonic acid”
Page 11 line 270. Remove “etc”.
Page 11 line 279 “removed the LcbHLHs with FPKM value less than 1”. Please specify if this was done considering <1 in all RNA-seq data sets or only in one. A TF could be specifically expressed in a genotype/condition.
Page 11 line 283. Move reference to the figure at the end of the sentence.
Page 14 line 335 “LcbHLH117/149 355 (MYC2 in Arabidopsis) could involve in the jasmonic acid (JA) ..” check grammar
Page 14 line 359 “Enhancer of Glabra 3 (EGL3).[38–41].” Remove “.”
Page 15 line 374 “and maize(n=208)[45] etc (The "n" here” delete “etc”
Page 15 lne 391 “including the synthesis of” change “synthesis” to “development”
Page 15 line 392. delete “etc”
Page 15 line 394 delete “and so on”
Page 15 line 399 “Many LcbHLHs performed differently in different tissues”. Change “performed into “are expressed”.
English is fine
Author Response
Dear Reviewer,
Thank you for your detailed review of our manuscript entitled “Genome-wide analysis of the bHLH gene family in Loropetalum chinense var. Rubrum: Identification, classification, evolution, and diversity of expression patterns under cultivation” (plants-2487725). The comments are of great help to improving the manuscript. We have studied the comments carefully and performed corresponding corrections in the revised manuscript. The point-by-point responses to the comments and suggestions are listed below.
We sincerely thank the editor and all reviewer for your valuable feedback that we have used to improve the quality of our manuscript. The reviewer comments are listed in normal black font and specific questions have been numbered. Our response is given in red font and the revisions of the manuscript are highlighted in yellow.
Suggestions for Authors
Major comments
In order to clarify the evolutionary relationship of L. chinense var. rubrum the authors construct a phylogenetic tree using Arabidopsis, why they did not include other species? this would provide a more accurate classification of the family members
Figure 4 needs some graphical improvements, as increase size of the motif legend, which is difficult to read in the current format. I would also suggest to use a more contrasting colour palette to represent the motifs in panel B. Also, why the motifs are not organised in progressive order in the legend? The legend needs to be revised: It seems that part of the description for panel C belongs to panel D.
The section on the expression data is the most interesting part of the manuscript and needs some revisions to be fully understood by the readers. For example, in Fig7 the material used for the trancriptomics should be better illustrated, the naming of the samples is not immediately clear to the reader, I suggest to extend the description as “XNFJF" could be “XNFJ-Flower" and “XNFJI” into “XNFJ-stageI. The text in panel B is too small, it is difficult to see the genotype labels. I suggest to use a vertical heat map instead of a circular one to better display the data. “XNFJI” appears twice, which makes difficult for the reader to interpret the data and discriminate between the two leaf.
At Page 12 line 325 “The results showed that these six genes were differentially up-regulated..” . This is not consistent with the data shown in figure 8. Where 3 genes show up-regulation and 3 down-regulation. That authors should revised the text accordingly. Finally, the authors should better explain why for one genotype a down-regulation of the candidate genes is shown and in the other an up-regulation is observed.
The paragraph 2.6. The protein-protein interaction network of candidate gene. need to be revised, the interaction found are not described in sufficient details. It seems that the in network is based on studies in Arabidopsis. What is the evidence for the conserve interaction? It is more a paragraph for the discussion section than a real result paragraph. I suggest to move the observation to the discussion this will help to put the finings in a broader context.
Minor comments
Page 1 line 24: “second-largest transcription factor in plants”. Add “family”
Page 3 line 105: delete “and so on”.
Page 3 lne 131 “Oryza sativa” change to italic
Page 5 line 152. remove “Arabidopssi thaliana” from the figure to legend, in Fig2 only bHLH proteins in L. chinense are shown.
page 10 line 247, delete “transcription factor complex (1) and RNA polymerase II transcription factor complex (6).” Most bHLH are transcription factors, it is expected that re part of TF complexes.
Page 10 line 256. Remove “etc”.
Page 10 line 261. Remove “and so on”.
Page 11 line 268. Change “MeJa” into “Jasmonic acid”
Page 11 line 270. Remove “etc”.
Page 11 line 279 “removed the LcbHLHs with FPKM value less than 1”. Please specify if this was done considering <1 in all RNA-seq data sets or only in one. A TF could be specifically expressed in a genotype/condition.
Page 11 line 283. Move reference to the figure at the end of the sentence.
Page 14 line 335 “LcbHLH117/149 355 (MYC2 in Arabidopsis) could involve in the jasmonic acid (JA) ..” check grammar
Page 14 line 359 “Enhancer of Glabra 3 (EGL3).[38–41].” Remove “.”
Page 15 line 374 “and maize(n=208)[45] etc (The "n" here” delete “etc”
Page 15 lne 391 “including the synthesis of” change “synthesis” to “development”
Page 15 line 392. delete “etc”
Page 15 line 394 delete “and so on”
Page 15 line 399 “Many LcbHLHs performed differently in different tissues”. Change “performed into “are expressed”.
Responses to reviewer (original comments by reviewer are in black color)
Q1: In order to clarify the evolutionary relationship of L. chinense var. rubrum the authors construct a phylogenetic tree using Arabidopsis, why they did not include other species? this would provide a more accurate classification of the family members
Reply: We think this is an excellent suggestion. We have modified the phylogenetic tree of Arabidopsis thaliana and Loropetalum chinense var.rubrum, and re-added the species of rice to support our classification criteria. According to the latest phylogenetic cluster analysis, we found that the subfamily classification of LcbHLHs did not change, but I mistakenly classified “AtbHLH158” in Orphans into XV and I has been corrected already. In addition, since the classification of peach (Prunus persica) species is also based on the classification criteria of Arabidopsis thaliana, the comparison between them is of little significance. What I needed was a more thorough study of the species on the bHLH classification criteria, so I replaced the peach (P persica) species in Table1 with Oryza sativa. The specific location is located in the yellow highlight part of the manuscript Page 6, Table 1 “Number of AtbHLHs ” and “Number of OsbHLHs ” .
Figure S1: Figure S1. Phylogenetic tree constructed with the sequences of the LcbHLHs, OsbHLHs, and AtbHLHs. (The red triangle represents AtbHLHs, the blue star represents LcbHLHs, and the green circle represents OsbHLHs.)
Q2: Figure 4 needs some graphical improvements, as increase size of the motif legend, which is difficult to read in the current format. I would also suggest to use a more contrasting colour palette to represent the motifs in panel B. Also, why the motifs are not organised in progressive order in the legend? The legend needs to be revised: It seems that part of the description for panel C belongs to panel D.
Reply: Thanks for your kindly suggestions. According to your suggestion, we have adjusted the Figure 4 accordingly. The size of the motif plate is appropriately increased and the motif label is added. In addition, we selected a more contrasting colour palette to show the motif and phylogenetic tree classification section for reading. I feel sorry for my carelessness. In the latest Figure 4, I have shown the legend part in a progressive order and adjusted the panels of B, C, and D. The modification part is located in Page 8 Line 203-208.
Figure 4. Phylogenetic relationships, conserved motifs, families of structural domains, and gene structure of the LcbHLHs. (A) Phylogenetic tree of LcbHLHs. (B) Conserved motifs of LcbHLHs. The 10 predicted motifs are depicted by rectangles of various colors. (C) The distribution of family domains and types of LcbHLHs. (D) Gene structure of LcbHLHs. The blue rectangle represents the exon, the orange rectangle represents the UTR, and the black line represents the intron. The size of these genes is indicated by the bottom coordinate.
Q3: The section on the expression data is the most interesting part of the manuscript and needs some revisions to be fully understood by the readers. For example, in Fig7 the material used for the trancriptomics should be better illustrated, the naming of the samples is not immediately clear to the reader, I suggest to extend the description as “XNFJF" could be “XNFJ-Flower" and “XNFJI” into “XNFJ-stageI. The text in panel B is too small, it is difficult to see the genotype labels. I suggest to use a vertical heat map instead of a circular one to better display the data. “XNFJI” appears twice, which makes difficult for the reader to interpret the data and discriminate between the two leaf.
Reply: We sincerely thank you for careful reading. As suggested by you, we adjusted the naming of all samples, such as “XNFJF” to “XNFJ-Flower”, “XNFJI” to “XNFJ-StageI”, and “XNFJII” to “XNFJ-StageII” (Figure 6). We corrected all “I-period/II-period” in the manuscript to “Stage-I/Stage-II” for readers to read (Page 10 Line290-291, Page 11 Line314-316, Line318-319, Line320 and Page12 Figure6 Line343-345, Figure7). The description of the transcriptome data sample is illustrated in the legend of figure 6, and we also explain in the materials and methods section (Page 16 Line 450-457). In order to see the labels of genotypes clearly, we transformed the circular heat map into a vertical heat map (Page 12 Figure 6). Due to the vertical heatmap clustering, the order of GroupIV and GroupV is reversed, and we have revised the corresponding text. (Page 10 Line 286-289).
Figure 6. Expression patterns of LcbHLHs (A) Four varieties with different colored leaves and flowers. Stage I and Stage II represent the first-stage and the second-stage leaves, respectively. The first stage is the young leaf stage; the second stage is the mature leaf period. (B) Expression patterns of the LcbHLHs. Expression profiles of LcbHLHs in different tissues. "XNFJ", "XNXY","XNXJ", and “XNNC” are the names of our selected materials.
Q4: At Page 12 line 325 “The results showed that these six genes were differentially up-regulated..” . This is not consistent with the data shown in figure 8. Where 3 genes show up-regulation and 3 down-regulation. That authors should revised the text accordingly. Finally, the authors should better explain why for one genotype a down-regulation of the candidate genes is shown and in the other an up-regulation is observed.
Reply: Thank you for your careful reading, I am extremely sorry for my negligence. I have already rewritten this part according to your suggestions. The six candidate genes were mainly obtained by phylogenetic analysis and transcriptome expression profile analysis. In previous studies, it has been shown that the IIIf family plays a positive regulatory role in the anthocyanin synthesis pathway. In our study, LcbHLH4/156/157 is located in the IIIf family and is highly expressed in the dark leaves (XNNC and XNFJ-StageI) of the transcriptome expression profile, suggesting that it positively regulates anthocyanin synthesis. However, LcbHLH114/117/149 is mainly selected by reading previous literature, through homologous comparison and based on the trend of transcriptome expression profile, its role in anthocyanin synthesis pathway is not yet clear. Therefore, we can clearly see that LcbHLH4/156/157, which are located in the IIIf family, are significantly up-regulated in mixed leaves (ML) and purple leaves(PL) in Figure 8, while the three candidate genes of LcbHLH114/117/149 are significantly down-regulated in purple leaves. This is because they mediate anthocyanin accumulation mainly through other pathways in the anthocyanin synthesis pathway, which is understandable. (The revised section is located at Page11 Line332-341).
Q5: The paragraph 2.6. The protein-protein interaction network of candidate gene. need to be revised, the interaction found are not described in sufficient details. It seems that the in network is based on studies in Arabidopsis. What is the evidence for the conserve interaction? It is more a paragraph for the discussion section than a real result paragraph. I suggest to move the observation to the discussion this will help to put the findings in a broader context.
Reply: We sincerely appreciate the valuable comments. We have adjusted the protein interaction network of six candidate genes. The interaction network of each candidate protein was listed one by one in order to describe the interaction between the candidate proteins in detail. In addition, we also mapped the common protein interaction network of the six genes, which is helpful for us to find out the relationship between them. We tried to construct protein interaction networks based on maize or rice databases, but found that the constructed interaction network annotations were often unclear, and some protein functions were not thoroughly studied than in Arabidopsis. Finally, as suggested by you, I transferred the results observed in the protein interaction network diagram to the discussion section and made a reasonable prediction in the discussion section. (Page14 Line358-374 Page15-16 Line427-444).
Figure 9. Interaction network for the six candidate LcbHLH proteins. The predicted results in parentheses are based on homologous genes in Arabidopsis. The colors of the lines represent the status of research into these protein interactions.
Q6: Page 1 line 24: “second-largest transcription factor in plants”. Add “family”
Reply: Thanks for the above suggestion. Based on your suggestion, we have added “family” in “second-largest transcription factor in plants”. (Page1 Line24)
Q7: Page 3 line 105: delete “and so on”.
Reply: Thanks for the above suggestion. Based on your suggestion, we have deleted “and so on” after “including LcCHS, LcDFR, LcCHI, LcANS”in Page 3 Line 105.
Q8: Page 3 line 131 “Oryza sativa” change to italic
Reply: Thanks for your careful checks, we feel sorry for our carelessness. We have changed “Oryza sativa” change to “Oryza sativa”. (Page 3 Line133)
Q9: Page 5 line 152. remove “Arabidopssi thaliana” from the figure to legend, in Fig2 only bHLH proteins in L. chinense are shown.
Reply: Thanks for your careful checks, we feel sorry for our carelessness. We have already removed “Arabidopssi thaliana” from the figure of legend. (Page 5 Line156-157)
Q10: page 10 line 247, delete “transcription factor complex (1) and RNA polymerase II transcription factor complex (6).” Most bHLH are transcription factors, it is expected that re part of TF complexes.
Reply: Thanks for the above suggestion. Based on your suggestion, we have deleted “transcription factor complex (1) and RNA polymerase II transcription factor complex (6).” (Page10 Line 253)
Q11: Page 10 line 256. Remove “etc”.
Reply: Thanks for the above suggestion. Based on your suggestion, we have removed “etc” after “like cold (7), iron (5) and light (2)”. (Page10 Line 262)
Q12: Page 10 line 261. Remove “and so on”.
Reply: Thanks for the above suggestion. Based on your suggestion, we have removed “and so on” after “including Box 4, Gap-box, G-Box, AE-box, I-box, GT1-motif, and TCCC-motif”. (Page10 Line 268)
Q13: Page 11 line 268. Change “MeJa” into “Jasmonic acid”
Reply: Thanks for the above suggestion. Based on your suggestion, we have Change “MeJa” into “Jasmonic acid” in Page 10 Line 276.
Q14: Page 11 line 270. Remove “etc”.
Reply: Thanks for the above suggestion. Based on your suggestion, we have removed “etc” after“and low temperature responsiveness (LTR)” in Page 10 Line 277.
Q15: Page 11 line 279 “removed the LcbHLHs with FPKM value less than 1”. Please specify if this was done considering <1 in all RNA-seq data sets or only in one. A TF could be specifically expressed in a genotype/condition.
Reply: Thank you for your careful reading, I am extremely sorry for my carelessness.
Page 10 line 279 “removed the LcbHLHs with FPKM value less than 1” mainly refers to the RNA-seq data of all samples. In all samples, FPKM value less than 1 is considered to be invalid expression. (Page 10 line 287)
Q16: Page 11 line 283. Move reference to the figure at the end of the sentence.
Reply: Thanks for the above suggestion. Based on your suggestion, we have moved reference to the figure 6B at the end of the sentence. (Page 10 Line 291)
Q17: Page 14 line 335 “LcbHLH117/149 355 (MYC2 in Arabidopsis) could involve in the jasmonic acid (JA) ..” check grammar
Reply: We are sorry for the grammatical problems and have correct them based your suggestions. We have rewritten the section of “2.6.The protein-protein interaction network of candidate gene” according to your suggestion, and the relevant section has been adjusted to “In addition, it is worth noting that LcbHLH117/149 (MYC2 in Arabidopsis) can also interact with JAZ proteins, indicating that LcbHLH117/ 149 are involved in the jasmonic acid (JA) signaling pathway….” (Page14 Line366-368)
Q18: Page 14 line 359 “Enhancer of Glabra 3 (EGL3).[38–41].” Remove “.”
Reply: Thank you for your careful reading. We have rewritten the section of “2.6.The protein-protein interaction network of candidate gene” according to your suggestion. We have deleted “Enhancer of Glabra 3 (EGL3)”, and this part is transferred to the discussion. (Page 15 Line 427-444)
Q19: Page 15 line 374 “and maize(n=208)[45] etc (The "n" here” delete “etc”
Reply: Thanks for the above suggestion. Based on your suggestion, we have removed “etc” after “and maize(n=208)” in Page15 Line 385.
Q20: Page 15 lne 391 “including the synthesis of” change “synthesis” to “development”
Reply: Thanks for the above suggestion. Based on your suggestion, we have changed “synthesis” into “development” in Page 15 Line 402-403.
Q21: Page 15 line 392. delete “etc”
Reply: Thanks for the above suggestion. Based on your suggestion, we have deleted “etc” after “and the response to various stresses” in Page 15 line 404.
Q22: Page 15 line 394 delete “and so on”
Reply: Thanks for the above suggestion. Based on your suggestion, we have deleted “and so on” after “such as gibberellin, jasmonic acid and auxin” in Page 15 line 405.
Q23: Page 15 line 399 “Many LcbHLHs performed differently in different tissues”. Change “performed into “are expressed”.
Reply: Thanks for the above suggestion. Based on your suggestion, we have changed “performed” into “are expressed” in Page 15 line 410
Once again, thank you very much for your comments and suggestions. A revised manuscript is attached. Should you have any questions, please contact us without any hesitation.
Sincerely yours,
Corresponding author
Yanlin Li
27th July, 2023

Reviewer 2 Report
Manuscript in written in popular form of language (remains secret... is speculated...) - check it and reformulate to scientific language.
Material and methods - prepare and use some schematic graphic visualization of individual steps of bioinformatic analysis, in this form, it is not comfortable to read it.
Dicsussion - be more specific about your own results and provide some hypothesis for further research.
Manuscript in written in popular form of language (remains secret... is speculated...) - check it and reformulate to scientific language.
Author Response
Dear Reviewer,
Thank you for your detailed review of our manuscript entitled “Genome-wide analysis of the bHLH gene family in Loropetalum chinense var. Rubrum: Identification, classification, evolution, and diversity of expression patterns under cultivation” (plants-2487725). The comments are of great help to improving the manuscript. We have studied the comments carefully and performed corresponding corrections in the revised manuscript. The point-by-point responses to the comments and suggestions are listed below.
We sincerely thank the editor and all reviewer for your valuable feedback that we have used to improve the quality of our manuscript. The reviewer comments are listed in normal black font and specific questions have been numbered. Our response is given in red font and the revisions of the manuscript are highlighted in green.
Suggestions for Authors
Manuscript in written in popular form of language (remains secret... is speculated...) - check it and reformulate to scientific language.
Material and methods - prepare and use some schematic graphic visualization of individual steps of bioinformatic analysis, in this form, it is not comfortable to read it.
Dicsussion - be more specific about your own results and provide some hypothesis for further research.
Comments on the Quality of English Language
Manuscript in written in popular form of language (remains secret... is speculated...) - check it and reformulate to scientific language.
Responses to reviewer (original comments by reviewer are in black color)
Q1:Manuscript in written in popular form of language (remains secret... is speculated...) - check it and reformulate to scientific language.
Reply: Thanks for your suggestion. We apologize for the poor language of our manuscript. We worked on the manuscript for a long time and asked professionals to polish the language in the revised manuscript. These changes will not influence the content and framework of the paper. Here we did not list the changes but marked in green in the revised paper. We appreciate for your warm work sincerely and hope that the correction will meet with approval.
Q2: Material and methods - prepare and use some schematic graphic visualization of individual steps of bioinformatic analysis, in this form, it is not comfortable to read it.
Reply: We think this is an excellent suggestion. I have showed the specific steps of the bHLH family identification in the form of a flow chart for better reading. Other biographical analysis is mainly formed directly through software or online websites.
(Page 16 line 468-477 Figure10)
Figure 10. Process of identifying the bHLH family proteins.
Q3: Dicsussion - be more specific about your own results and provide some hypothesis for further research.
Reply: We sincerely appreciate the valuable comments. We have carefully discussed the content of the analysis. By referring to the previous literature, the analysis results are summarized and discussed, and reasonable assumptions are further put forward in the discussion part. In Section 3.1, we compared the number of identified LcbHLHs and the distribution of subfamily members with previous studies in other species to explore the evolution of LcbHLHs. These results give us a certain understanding of the evolution and classification of LcbHLHs in L chinense. (Page 15 line385-389 line394-398). In addition, we conducted evolutionary analysis, GO annotation, homeostatic element analysis, and transcriptome expression profile analysis to have a certain understanding of the function of LcbHLHs in the discussion section 3.2. Six candidate genes that may be involved in anthocyanin synthesis were selected by comprehensive analysis. Through literature reading and protein interaction network, there is a bold conjecture on the action pathway of these six genes—these six mainly affect anthocyanin synthesis through two pathways.(Page 15 line 437-441).
Once again, thank you very much for your comments and suggestions. A revised manuscript is attached. Should you have any questions, please contact us without any hesitation.
Sincerely yours,
Corresponding author
Yanlin Li
27th July, 2023

Reviewer 3 Report
Suggestions,
1) The authors have performed in silico analyses. Such studies often need to be validated through wet laboratory experiments. I suggest functional characterization of 1-2 interesting genes (over-expression or mutant lines). This will add some value to paper.
2) Moderate editing of English language is needed.
3) What is applicability of this research? Which research question is addressed here? Include detailed information.
4) Figure 4 is not clear. Provide high quality figure.
Moderate editing of English language is needed.
Author Response
Dear Reviewer,
Thank you for your detailed review of our manuscript entitled “Genome-wide analysis of the bHLH gene family in Loropetalum chinense var. Rubrum: Identification, classification, evolution, and diversity of expression patterns under cultivation” (plants-2487725). The comments are of great help to improving the manuscript. We have studied the comments carefully and performed corresponding corrections in the revised manuscript. The point-by-point responses to the comments and suggestions are listed below.
We sincerely thank the editor and all reviewer for your valuable feedback that we have used to improve the quality of our manuscript. The reviewer comments are listed in normal black font and specific questions have been numbered. Our response is given in red font and the revisions of the manuscript are highlighted in blue.
Suggestions for Authors
1) The authors have performed in silico analyses. Such studies often need to be validated through wet laboratory experiments. I suggest functional characterization of 1-2 interesting genes (over-expression or mutant lines). This will add some value to paper.
2) Moderate editing of English language is needed.
3) What is applicability of this research? Which research question is addressed here? Include detailed information.
4) Figure 4 is not clear. Provide high quality figure.
Comments on the Quality of English Language
Moderate editing of English language is needed.
Responses to reviewer (original comments by reviewer are in black color)
Q1: The authors have performed in silico analyses. Such studies often need to be validated through wet laboratory experiments. I suggest functional characterization of 1-2 interesting genes (over-expression or mutant lines). This will add some value to paper.
Reply: Thank you very much for pointing out this important issue. We agree with your comments that preliminary functional verification experiments are also necessary. However, due to the time and technical limitations, it is difficult to supplement the experimental verification for the time being. The genetic transformation experiment conducted a long time and the smoothness of the experiment cannot be guaranteed. But we will still work in this direction, thank you for your advice to provide the direction for our next research.
In this study, we aimed to explore the bHLH regulatory mechanism of anthocyanin synthesis in Loropetalum chinense var.rubrum, which mainly about the molecular mechanism, so we focused on the results of bioinformatics analysis. Through the analysis of the bHLH gene family of Loropetalum chinense var.rubrum, the LcbHLHs related to anthocyanin synthesis were screened out, and it will provide a solid theoretical support for the subsequent molecular mechanism of leaf color regulation of Loropetalum chinense var.rubrum.
Q2: Moderate editing of English language is needed.
Reply: We apologize for the poor language of our manuscript. We worked on the manuscript for a long time and the repeated addition and removal of sentences obviously led to poor readability. We have now worked on both language and readability in manuscript, we also have invited professionals to polish the language. We really hope that the flow and language level have been substantially improved.
Q3: What is applicability of this research? Which research question is addressed here? Include detailed information.
Reply: This study is mainly applicable to molecular breeding. Loropetalum chinensis var.rubrum is a typical flower and leaf ornamental plant in Asia, Europe and America. It has high ornamental value and economic value, and its leaf color often affects the ornamental value of Loropetalum chinensis var.rubrum. One of the essential regulatory gene in the anthocyanin synthesis pathway is bHLH. Here, the bioinformatics analysis and expression analysis of LcbHLHs in Loropetalum chinense var.rubrum were carried out, and the candidate LcbHLHs related to the regulation of anthocyanin synthesis were purposefully screened out, and the interaction network between candidate proteins was further predicted. These analysis results provided theoretical support for our subsequent research on the molecular mechanism of leaf color regulation in Loropetalum chinense var.rubrum and were conducive to improving plants at the molecular level for cultivating new varieties with higher ornamental value.(Page 3 line114-119)
Q4: Figure 4 is not clear. Provide high quality figure.
Reply: Thanks for your kindly suggestions. According to your suggestion, we have adjusted the Figure 4 accordingly and provided a clearer picture. The size of the motif plate is appropriately increased and the motif label is added. In addition, we selected a more contrasting colour palette to show the motif and phylogenetic tree classification section for reading. (Page 8 Figure 4)
Q5: Comments on the Quality of English Language
Moderate editing of English language is needed.
Reply: We apologize for the poor language of our manuscript again. We worked on the manuscript for a long time and the repeated addition and removal of sentences obviously led to poor readability. We have now worked on both language and readability in manuscript, we also have invited professionals to polish the language. We really hope that the flow and language level have been substantially improved.
Once again, thank you very much for your comments and suggestions. A revised manuscript is attached. Should you have any questions, please contact us without any hesitation.
Sincerely yours,
Corresponding author
Yanlin Li
27th July, 2023

Round 2
Reviewer 2 Report
Dear authors. The manuscript was improved substantialy. I have no further suggestions.
Reviewer 3 Report
The author has not adequately addressed all comments. In my opinion, the revision did not improve the publication. It still seems to be a rough draft. English is still a concern. Major edition in English is needed.
My suggestion is to resubmit after making drastic changes.
English is still a concern. Major edition in English is needed.